# FGF signalling controls the specification of hair placode-derived SOX9 positive progenitors to Merkel cells

Minh Binh Nguyen[1], Idan Cohen[1], Vinod Kumar[2], Zijian Xu[3], Carmit Bar[1], Katherine L. Dauber-Decker[1], Pai-Chi Tsai[4], Pauline Marangoni[5], Ophir D. Klein [5,6], Ya-Chieh Hsu[4], Ting Chen[3], Marja L. Mikkola[2] & Elena Ezhkova [1]

Merkel cells are innervated mechanosensory cells responsible for light-touch sensations. In murine dorsal skin, Merkel cells are located in touch domes and found in the epidermis around primary hairs. While it has been shown that Merkel cells are skin epithelial cells, the progenitor cell population that gives rise to these cells is unknown. Here, we show that during embryogenesis, SOX9-positive (+) cells inside hair follicles, which were previously known to give rise to hair follicle stem cells (HFSCs) and cells of the hair follicle lineage, can also give rise to Merkel Cells. Interestingly, while SOX9 is critical for HFSC specification, it is dispensable for Merkel cell formation. Conversely, FGFR2 is required for Merkel cell formation but is dispensable for HFSCs. Together, our studies uncover SOX9(+) cells as precursors of Merkel cells and show the requirement for FGFR2-mediated epithelial signalling in Merkel cell specification.

[1] Black Family Stem Cell Institute, Department of Cell, Developmental, and Regenerative Biology, Icahn School of Medicine at Mount Sinai, 1 Gustave L. Levy Place, New York, NY 10029, USA. [2] Developmental Biology Program, Institute of Biotechnology, University of Helsinki, 00014 Helsinki, Finland. [3] National Institute of Biological Sciences, Beijing 102206, China. [4] Department of Stem Cell and Regenerative Biology, Harvard University, Harvard Stem Cell Institute, Cambridge, MA 02138, USA. [5] Department of Orofacial Sciences and Program in Craniofacial Biology, University of California San Francisco, San Francisco, CA 94143, USA. [6] Department of Pediatrics and Institute for Human Genetics, University of California San Francisco, San Francisco, CA 94143, USA. These authors contributed equally: Minh Binh Nguyen, Idan Cohen. Correspondence and requests for materials should be addressed to E.E. (email: elena.ezhkova@mssm.edu)

The skin epithelium is an essential barrier that protects the body from the environment, helps to maintain temperature and keep water within the body, and performs sensory functions[1]. These activities are largely provided by the epidermis, hair follicles, and specialized cells, including Merkel cells, which respectively serve protective barrier functions, provide thermo-protection, and are involved in mechanosensation[1–3]. While much has been learned about the development of hair follicles and the epidermis, the processes controlling the specification of Merkel cell are largely understudied. These mechanosensory cells are innervated by afferent neurons and are responsible for the tactile discrimination of the shape and texture of objects[4,5]. Recent studies have shown that upon touch stimulation, Merkel cells produce ionic currents that induce a release of neuro-transmitters, which trigger firing of the afferent neurons that innervate Merkel cells[6–8]. Moreover, mice without Merkel cells are unable to discriminate between different textures when performing behavioural tasks[9].

Much of what we know about the biology of Merkel cells came from studies of murine back skin, where Merkel cells are located in specialized structures called touch domes[10]. Touch domes consist of Merkel cells, specialized keratinocytes, and afferent neurons, and are located exclusively around primary hair follicles, which represent 1–3% of the mouse hair coat[3,5,6,9,11]. Although Merkel cells were discovered more than 100 years ago, their cell of origin is still unknown. It was long believed that Merkel cells originate from the neural crest[12] until fate-mapping experiments showed that embryonic epidermal progenitor cells that express keratin (KRT) 14 give rise to Merkel cells[13–15]. While these studies showed that Merkel cells are of skin epithelial origin, they also raised questions as to whether a specific population of Merkel cell precursors exists. Indeed, at embryonic day (E) 14.5 when the first Merkel cells appear, embryonic epidermal progenitors are no longer a single layer of cells, as epidermal stratification has initiated and hair follicles are at the placode stage[13,16,17].

In this study, we analysed the appearance of the first Merkel cells in the skin during embryogenesis and found that these cells appear inside of developing hair follicles. By performing lineage tracing experiments, we discovered that SOX9(+) cells, which in prior literature have been proven to give rise to cells of the hair follicle lineage, including HFSCs that maintain postnatal hair follicle growth and homoeostasis, can also give rise to Merkel Cells. We dissected the molecular mechanisms controlling the specification of SOX9(+) cells to Merkel cells and showed that although SOX9 is critical for SOX9(+) cell specification to HFSCs, it is dispensable for Merkel cell formation. Interestingly, FGFR2-mediated signalling in the skin epithelium is critical for Merkel cell development but is not required for HFSC specification. Taken together, our studies uncovered that SOX9(+) cells located within the developing hair placodes give rise to Merkel cells through FGFR2-mediated signalling.

## Results

**Merkel cells form inside hair placodes during development.** To gain insights into the cell of origin of Merkel cells, we aimed to visualize where Merkel cells appear in embryonic skin. ATOH1 is one of the earliest Merkel cell differentiation markers[16], and thus we set out to determine where ATOH1(+) cells first appear in the skin. We crossed *Atoh1-GFP* mice, which contain an enhanced green fluorescent protein (GFP) fused to the 3′-end of the atonal homologue 1 gene (*Atoh1*), with *R26-mT/mG* mice and collected embryos at E15. By performing confocal imaging we demonstrated that ATOH1-GFP(+) Merkel cells were not present in the basal layer of the epidermis (Fig. 1a, left), and instead they were found within primary hair placodes (Fig. 1a, right). These data are

consistent with previous studies of KRT8, an early Merkel cell marker which appears after ATOH1 induction, showing that KRT8(+) cells are present inside of developing hair follicles at E15[18].

The finding that Merkel cells are formed inside of developing hair follicles prompted us to investigate whether a cell population located in hair follicles gives rise to Merkel cells. At the time when the first Merkel cells appear, hair follicle lineage specification has already occurred. The hair follicles contain discrete cell populations with different locations within hair placodes[19]. Cells at the leading edge of developing hair follicles express the transcription factor LHX2, whereas suprabasal hair placode cells are positive for the stem cell pioneer factor SOX9 (Fig. 1b–g)[20–22]. We performed immunofluorescence analysis to determine whether the first Merkel cells appear next to SOX9(+) or LHX2(+) hair follicle cells. At E14.5, when the first Merkel cells are detected, ATOH1-GFP(+) cells were located in hair placodes, near SOX9 (+) cells, and further away from LHX2(+) cells (Fig. 1c, f). The proximity of Merkel cells to SOX9(+) cells was even more apparent at E15.5, when more Merkel cells had developed (Fig. 1d, g). Regardless of cell proximity, ATOH1-GFP(+) Merkel cells did not co-label with SOX9 (Supplementary Fig. 1a–c) or LHX2 (Fig. 1f, g). We thus concluded that during development, Merkel cells originate inside of hair placodes and are located near the SOX9(+) cell population.

**SOX9(+) cells give rise to Merkel cells in hair placode.** The emergence of Merkel cells inside hair follicles prompted us to hypothesize that one of the cell populations inside developing hair follicles is a precursor population for Merkel cells. To test this hypothesis, we used *Sox9-CreER*; *R26-mT/mG* and *Lhx2-CreER*; *R26-tdTomato* mice to fate map cells originating from SOX9(+) and LHX2(+) cells, correspondingly. Embryos were treated with Tamoxifen at E13.5–E14.5, at which time SOX9(+) and LHX2(+) cells are present but Merkel cells have not yet been specified, and collected from both lines at postnatal day (P) 0 (Fig. 2a) and at E16 for *Sox9-CreER*; *R26-mT/mG* (Supplementary Fig. 1d).

Consistent with previous findings[20], our analysis of E16 and P0 *Sox9-CreER*; *R26-mT/mG* mice revealed the presence of GFP(+) cells in the hair follicle outer root sheath (Fig. 2b and Supplementary Fig. 1e). Interestingly, immunofluorescence analysis revealed that GFP staining was found in more than 90% of KRT8(+) Merkel cells (Fig. 2b, c and Supplementary Fig. 1d, e). While we observed that roughly 10% of KRT8(+) cells were not GFP-labelled (Fig. 2b, c and Supplementary Fig. 1d, e), this is likely due to incomplete recombination, as a similar percentage of SOX9(+) cells remained GFP-negative (Supplementary Fig. 1f, g). In contrast, immunofluorescence analysis of tdTOMATO in P0 *Lhx2-CreER*; *R26-tdTomato* skins revealed that only 22% of KRT8 (+) Merkel cells were tdTOMATO(+) (Fig. 2d, e). These fate-mapping results thus show that SOX9(+) rather than LHX2(+) cells preferentially give rise to Merkel cells.

**SHH signalling promotes the specification of SOX9(+) cells.** SHH signalling has been shown to be both necessary and sufficient for Merkel cell formation[23,24]. We thus asked if SHH controls the Merkel cell lineage by regulating the establishment of SOX9(+) cells. To test this hypothesis, we first overexpressed SHH in the embryonic epidermis by injecting high-titre lentiviruses expressing a doxycycline-inducible *Shh-PGK-H2B-RFP* construct into the amniotic fluid of E9 *Rosa26-rtTA* embryos to infect the epidermis[23,25]. We administered doxycycline to the pregnant dams starting at E12 and the infected embryos were collected at E17 (Supplementary Fig. 2a). Overexpression of SHH

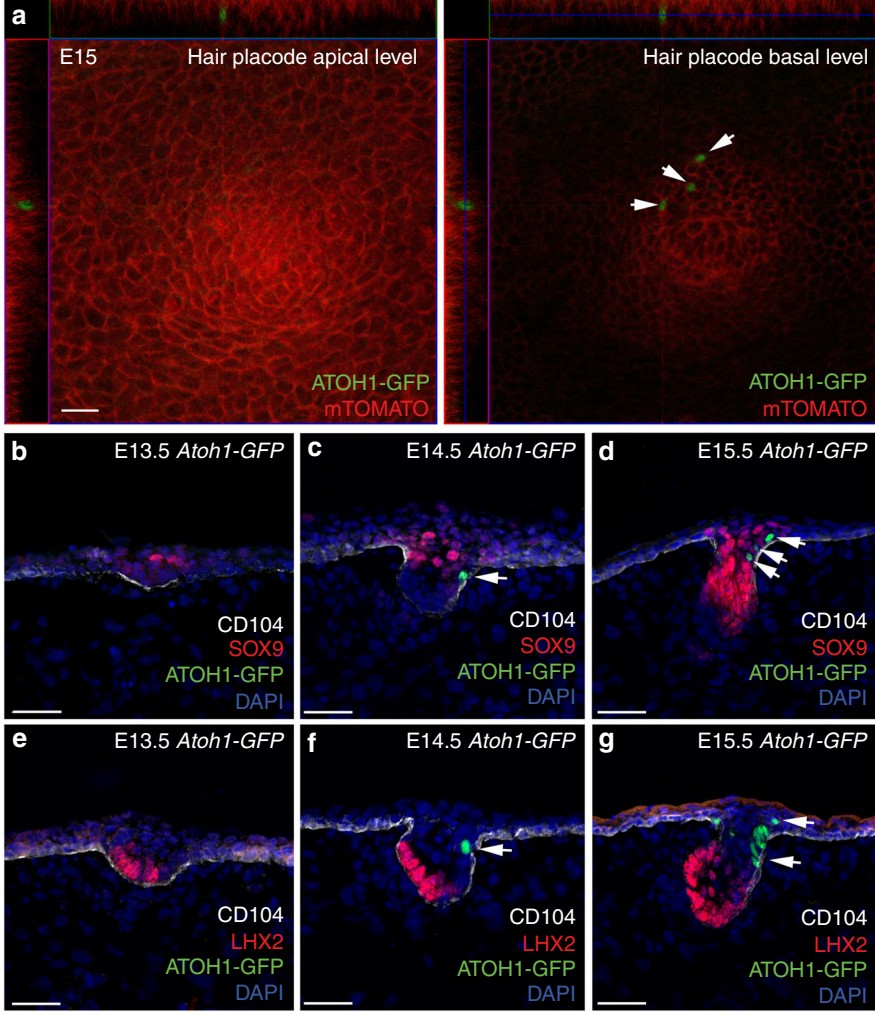

**Fig. 1** Merkel cells are specified from cells inside the hair placode. **a** Immunofluorescence analysis of E15 ATOH1-GFP skins shows the localization of ATOH1-GFP(+) early Merkel cells inside hair placodes. Scale = 20 μm. **b-d** Immunofluorescence staining for the hair follicle stem cell marker SOX9 (red) and early Merkel cell specification marker ATOH1-GFP (green) during early hair follicle morphogenesis at E13.5 (**b**), E14.5 (**c**), and E15.5 (**d**). Basement membrane is labelled by CD104. White arrow indicates Merkel cells. Note that ATOH1-GFP(+) Merkel cells are located inside the hair follicle, adjacent to SOX9(+) cells. **e-g** Immunofluorescence staining for LHX2 (red) and ATOH1-GFP (green) during early hair follicle morphogenesis at E13.5 (**e**), E14.5 (**f**), and E15.5 (**g**). Basement membrane is labelled by CD104. White arrow indicates Merkel cells. Scale = 50 μm in all panels

resulted in ectopic KRT8(+) Merkel cells that were located in areas with SHH overexpression, identified by RFP expression (Supplementary Fig. 2c), whereas in control non-infected areas, Merkel cells were present only near primary hairs (Supplementary Fig. 2b). Interestingly, immunostaining for SOX9 showed that the same SHH-expressing areas had ectopic SOX9(+) cells present in the epidermis (Supplementary Fig. 2c), whereas in non-infected control areas, SOX9(+) cells were found only inside hair follicles (Supplementary Fig. 2b).

We next analysed whether SHH signalling is required for the specification of SOX9(+) cells by deleting the essential SHH-signalling signal transducer Smoothened[23,26]. We generated skin epithelium-conditional knockout mice of Smoothened (Smo cKO), by crossing Smo floxed mice with mice expressing Krt14-Cre, which is active in embryonic epidermal progenitors starting at E12 (ref. [27]). Consistent with previous reports[23,24], there was a dramatic reduction in the number of Merkel cells in P0 Smo cKO skin compared to control skin (Supplementary Fig. 2d). Importantly, immunofluorescence analysis of SOX9 showed that there was an absence of SOX9(+) cells in embryonic and neonatal hair

follicles of Smo-null skin, whereas these cells were apparent in control hair follicles (Supplementary Fig. 2d, e). Together, these results showed that in the skin epithelium, SHH signalling promotes specification of SOX9(+) cells and Merkel cell formation.

**SOX9 and NFATc1 are dispensable for Merkel cell formation.** SOX9(+) cells give rise to HFSCs that, in adulthood, continuously fuel the regeneration of hair follicles during each hair cycle[20,28]. Importantly, transcription factor SOX9 is required for the specification of SOX9(+) cells to HFSCs[20,29]. As SOX9(+) cells also give rise to Merkel cells, we asked if SOX9 is required for Merkel cell specification. To test this, we generated skin epithelium-conditional knockout mice of Sox9 (Sox9 cKO) by crossing Sox9 floxed mice with Krt14-Cre mice. Immunofluorescence analysis of Sox9-null skins confirmed the loss of SOX9 in hair follicles (Supplementary Fig. 3a). However, despite the loss of SOX9 in the skin epithelium, immunolabelling for early and late Merkel cell differentiation markers, KRT8 and

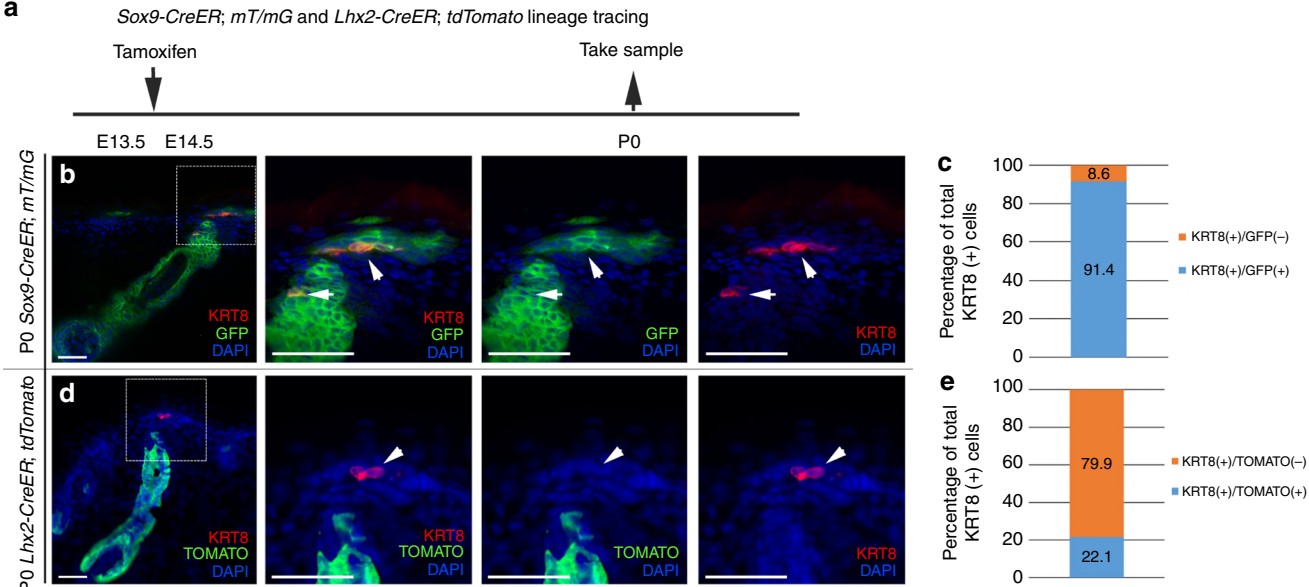

**Fig. 2** Merkel cells originate from SOX9-expressing hair follicle cells during embryogenesis. **a** Experimental design for *Sox9-CreER; mT/mG* and *Lhx2-CreER; tdTomato* lineage tracing. **b** *Sox9-CreER; mT/mG* lineage tracing. SOX9-traced cells are GFP(+) (green). KRT8(+) Merkel cells are stained in red. Right side panels show zoom in images. White arrows are pointing to Merkel cells. Note the overlap between SOX9 progenies and KRT8 staining. **c** Quantification of Merkel cells positive for KRT8 and GFP in *Sox9-CreER; mT/mG* skin sections at P0. *n* = 4. Data are presented as a percentage of the total number of cells quantified. Bar graph shows the percentage of KRT8(+) cells which are co-labelled with GFP (blue) or not co-labelled with GFP (orange). **d** *Lhx2-CreER; tdTomato* lineage tracing. LHX2-traced cells are tdTOMATO(+) (green). KRT8(+) Merkel cells are stained in red. Right side panels show zoom in images. White arrows are pointing to Merkel cells. **e** Quantification of Merkel cells positive for KRT8 and tdTOMATO in *Lhx2-CreER; tdTomato* skin sections at P0. *n* = 4. Data are presented as a percentage of the total number of cells quantified. Bar graph shows the percentage of KRT8(+) cells which are co-labelled with tdTOMATO (blue) or not co-labelled with tdTOMATO (orange). Scale = 50 μm in **b** and **d**

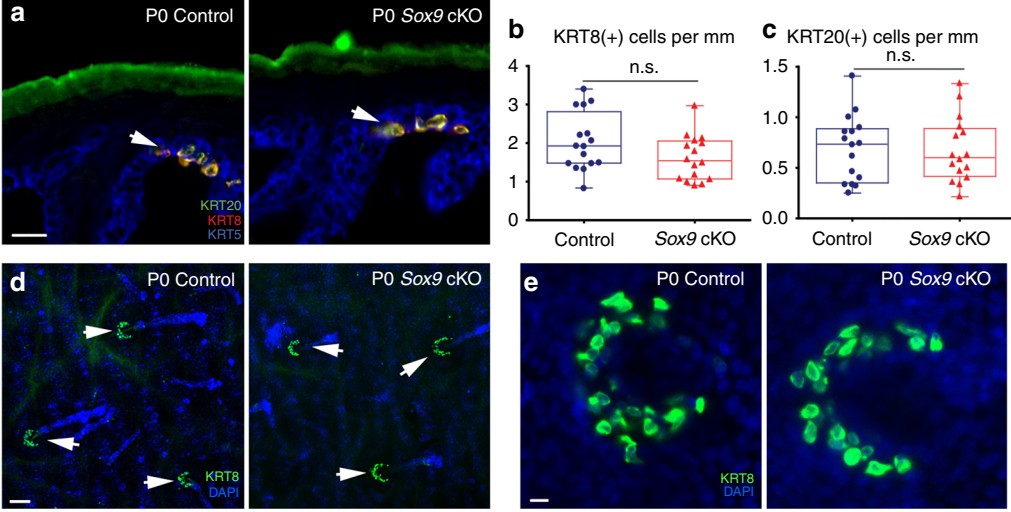

**Fig. 3** Transcription factor SOX9 is dispensable for Merkel cell formation. **a** Immunofluorescence staining for Merkel cell markers KRT8 (red) and KRT20 (green) in *Sox9* cKO and control back skin. Basal layer in stained with KRT5 (blue). White arrows indicate Merkel Cells. **b** Quantification of KRT8(+) Merkel cells in *Sox9* cKO and control back skins, *p* = 0.1268. *n* = 3. **c** Quantification of KRT20(+) Merkel cells in *Sox9* cKO and control back skins, *p* = 0.9549. *n* = 3. **d, e** Whole-mount immunofluorescence staining for KRT8 in *Sox9* cKO and control back skins. White arrows are pointing to touch domes. Scale = 20 μm in **a, e**. Scale = 100 μm in **d**. The data presented in box plots (**c, d**) shows the median with 25th and 75th percentile borders. Whiskers extend from minimum to maximum. n.s., not significant (the Mann–Whitney test)

KRT20, respectively, demonstrated no significant changes in the number of KRT8(+) and KRT20(+) cells (Fig. 3a–e).

Another HFSC marker, NFATc1, starts being expressed in SOX9 (+) cells in the subsequent hair peg stage, and NFATc1 is critical for acquiring the quiescent state of HFSCs[20,30]. To test if NFATc1 is required for Merkel cell specification, we generated and analysed skin epithelium-conditional knockout mice of *Nfatc1* (*Nfatc1* cKO)

by crossing *Nfatc1* floxed mice with *Krt14-Cre* mice. As with *Sox9* cKO mice, immunofluorescence analysis of Merkel cell markers KRT8 and KRT20 showed that the number of Merkel cells was not changed between control and *Nfatc1* cKO skins (Supplementary Fig. 3b–d). Therefore, while SOX9 and NFATc1 are important for HFSC specification and quiescence, respectively, they are dispensable for Merkel cell formation.

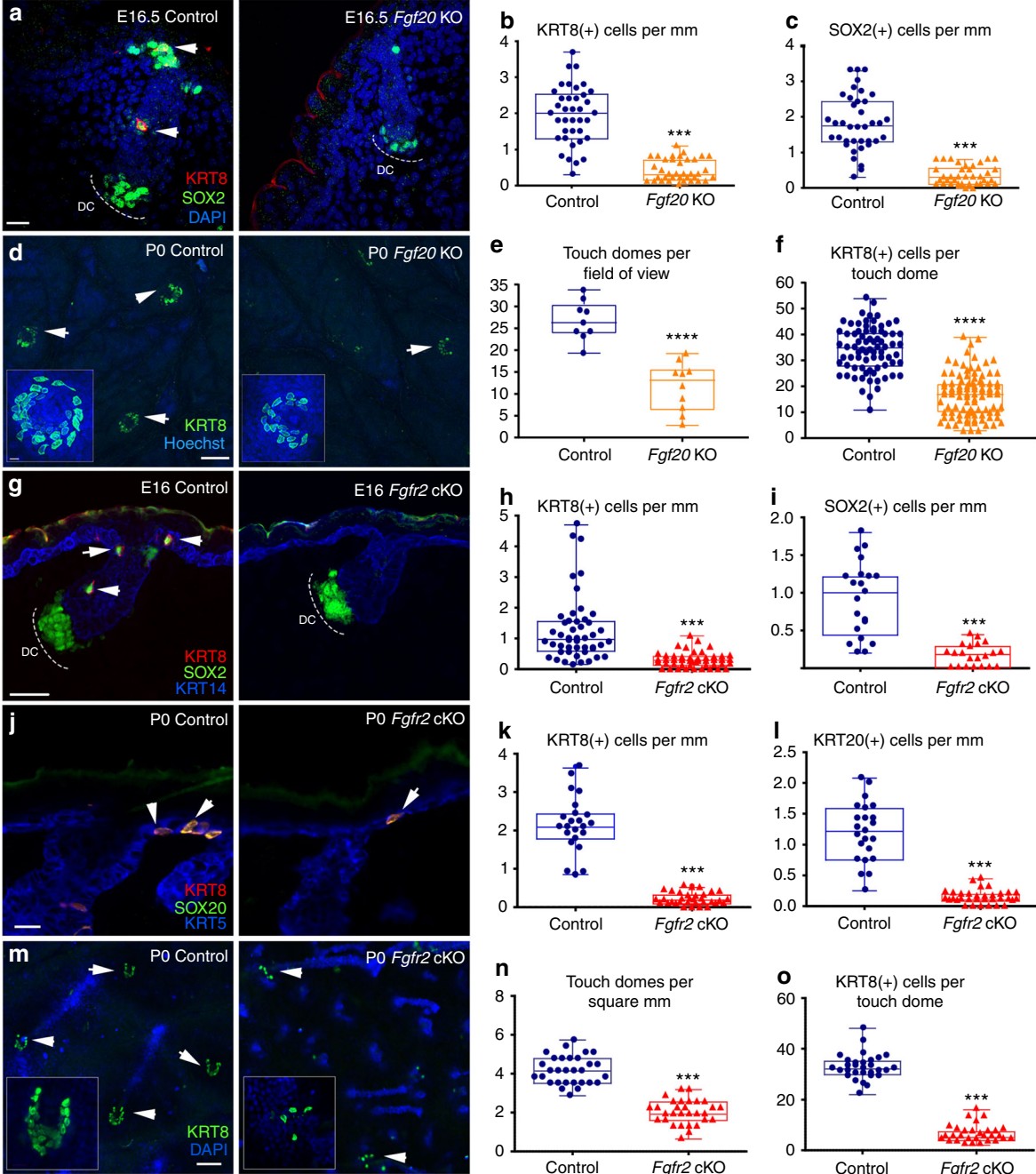

**Fig. 4** FGFR2 controls Merkel cell specification. **a** Immunofluorescence staining for KRT8 (red) and SOX2 (green) in *Fgf20* KO and control back skins. **b** Quantification of KRT8(+) Merkel cells in *Fgf20* KO and control back skins at E16.5, *p* < 0.001. *n* = 3. **c** Quantification of SOX2(+) Merkel cells in *Fgf20* KO and control back skins at E16.5, *p* < 0.001. *n* = 3. **d** Whole-mount immunofluorescence staining for KRT8 in *Fgf20* KO and control back skins. **e** Quantification of touch domes in *Fgf20* KO and control back skins at P0, *p* < 0.001. *n* = 9. **f** Quantification of KRT8(+) Merkel cells per Touch Dome in *Fgf20* KO and control back skins at P0, *p* < 0.001. *n* = 9. **g** Immunofluorescence staining for KRT8 (red) and SOX2 (green) in *Fgfr2* cKO and control back skins at E16. **h** Quantification of KRT8(+) Merkel cells in *Fgfr2* cKO and control back skins at E16, *p* < 0.001. *n* = 3. **i** Quantification of SOX2(+) Merkel cells in *Fgfr2* cKO and control back skins at E16, *p* < 0.001. *n* = 3. **j** Immunofluorescence staining for KRT8 (red) and KRT20 (green) in *Fgfr2* cKO and control back skins at P0. **k** Quantification of KRT8(+) Merkel cells in *Fgfr2* cKO and control back skins at P0, *p* < 0.001. *n* = 3. **l** Quantification of KRT20(+) Merkel cells in *Fgfr2* cKO and control back skins at P0, *p* < 0.001. *n* = 3. **m** Whole-mount immunofluorescence staining for KRT8 in *Fgfr2* cKO and control back skins. **n** Quantification of touch domes (TD) in *Fgfr2* cKO and control back skins at P0, *p* < 0.001. *n* = 4. **o** Quantification of KRT8(+) Merkel cells per TD in *Fgfr2* cKO and control back skins at P0, *p* < 0.001. *n* = 4. White arrows indicate Merkel cells and dashed lines indicate location of Dermal Condensates (DC) (**a**, **g**, **j**). Scale = 20 μm in **a**, **j**. Scale = 50 μm in **g**. White arrows are pointing to touch domes (**d**, **m**). Scale = 100 μm in **d**, **m**. The data presented in box plots (**b**, **c**, **e**, **f**, **h**, **i**, **k**, **l**, **n**, **o**) shows the median with 25th and 75th percentile borders. Whiskers extend from minimum to maximum. ***p* < 0.001 (the Mann–Whitney test)

**Merkel cell specification requires FGFR2-mediated signalling.** While SOX9(+) cells have been shown to give rise to the hair follicle lineage, our studies now unveil that SOX9(+) cells also give rise to Merkel cells. We next aimed to uncover the molecular processes that control the specification of SOX9(+) cells to the Merkel cell lineage. Fibroblast growth factor (FGF) 20 is expressed by hair placodes (Supplementary Fig. 4a and Supplementary Fig. 4b, left) and is required for the proper development of primary follicles[31]. Interestingly, transcriptional analysis of *Fgf20*-null skins revealed that at E15.5, there is decreased expression of the early Merkel cell differentiation gene, *Atoh1*, in *Fgf20*-null skin compared to control[24]. To determine if decreased expression of *Atoh1* results in changes in Merkel cell numbers, we performed immunofluorescence analysis of Merkel cell markers SOX2 and KRT8 in *Fgf20*$^{\beta\text{-gal}/\beta\text{-gal}}$ (*Fgf20*-null) skins, followed by quantification. Indeed, we observed a significant reduction in the number of KRT8(+) and SOX2(+) cells in *Fgf20*-null skins compared to controls at E16 (Fig. 4a–c) and at P0 (Fig. 4d–f). FGF10, another FGF ligand, is also highly expressed in the developing skin, but is localized in dermal condensates instead of hair placodes (Supplementary Fig. 4b, right)[32]. Interestingly, and in contrast to *Fgf20*-null skin analysis, immunofluorescence analysis of KRT8 did not reveal any differences in the number of Merkel cells in the skins of *Fgf10*-null mice compared to controls (Supplementary Fig. 4c, d).

During primary hair formation, FGF20 functions in both the epithelium and the mesenchyme[33,34]. We thus asked whether FGF signalling functions directly in the skin epithelium to specify Merkel cells, or indirectly through control of the mesenchyme. Analysis of published RNA-seq data[35][http://hair-gel.net] revealed that among four FGF receptors (FGFR), only FGFR2 is expressed at high levels in E14.5 hair placodes, whereas FGFR1

and FGFR2 are both expressed in the dermal condensate, which consists of dermal papilla precursor cells (Supplementary Fig. 4e). Importantly, immunofluorescence analysis using *Fgfr2-mCherry* reporter mice demonstrated that FGFR2 expression overlaps with SOX9(+) cells and KRT8(+) Merkel cells at E16 (Supplementary Fig. 4f, g). Furthermore, by analysing the skins of E16 *Fgf20*$^{\beta\text{-gal}}$ mice, we observed a gradient in the expression of FGF20$^{\beta\text{-gal}}$ reporter in FGFR2(+) and SOX9(+) cells, where some FGFR2 (+) and SOX9(+) cells had some levels of FGF20$^{\beta\text{-gal}}$ reporter, while others had none (Supplementary Fig. 4h, i).

To test whether transduction of FGF signalling in skin epithelial cells controls Merkel cell formation, we generated skin epithelium-conditional knockout mice of *Fgfr2* by crossing *Fgfr2* floxed mice with *Krt14-Cre* mice. Because there are several FGFR2 isoforms, we generated a conditional deletion of exon 5 of *Fgfr2*, which is common to all isoforms, thus abrogating FGFR2 function, as previously described[36]. Immunofluorescence analysis of E16 embryos confirmed efficient loss of FGFR2 protein from the skin epithelium of *Fgfr2* cKO mice compared to controls (Supplementary Fig. 4j). Analysis of E16 *Fgfr2* cKO skins revealed significant reductions in the numbers KRT8(+) and SOX2(+) Merkel cells (Fig. 4g–i). SOX2 staining was observed in the dermal condensates of control and *Fgfr2*-null hair follicles, indicating that epithelial FGFR2 was not essential for dermal condensate formation (Fig. 4g). The reduced number of Merkel cells observed in E16 *Fgfr2* cKO skins was not due to apoptotic cell death, as assessed by TUNEL staining (Supplementary Fig. 4l). Reduced number of Merkel cells persisted into neonatal life, as shown by immunofluorescence analyses of KRT8 and KRT20 in P0 control and *Fgfr2* cKO mice (Fig. 4j–o). Thus, FGFR2-mediated signalling in the skin epithelium is required for Merkel cell formation.

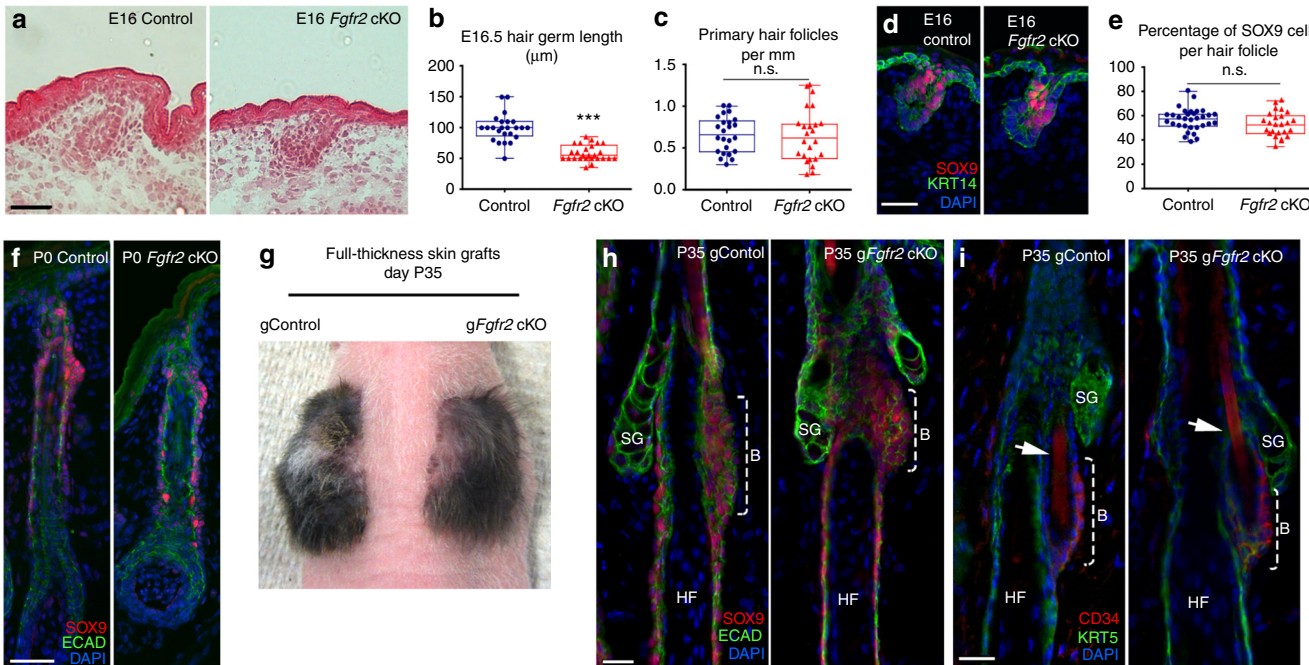

**Fig. 5** FGFR2 is not required for HFSC specification. **a** Hematoxylin and eosin (H&E) analysis of E16 skins of *Fgfr2* cKO and control mice. **b** Quantification of hair follicle length, *p* < 0.001. *n* = 4. **c** Quantification of primary hair follicles number per mm of back skin at E16, *p* = 0.5358. *n* = 4. **d** Immunofluorescence staining for SOX9 (red) at E16. **e** Quantifications of the percentage of SOX9(+) cells per hair follicle in control and *Fgfr2* cKO skins at E16, *p* = 0.1570. *n* = 4. **f** Immunofluorescence staining for SOX9 (red) at P0. **g** Appearance of full-thickness skin engraftment of P0 skins onto *Nude* mice 35 days post-engraftment (P35). Grafted *Fgfr2* cKO, g*Fgfr2* cKO; grafted control, gControl. **h** Immunofluorescence staining for HFSC marker SOX9 (red) in P35 g*Fgfr2* cKO and gControl. **i** Immunofluorescence staining for HFSC marker CD34 (red) in P35 g*Fgfr2* cKO and gControl skins. Arrow points to a club hair. Scale = 50 μm in **a**, **d**, **f**. Scale = 20 μm in **h**, **i**. The data presented in box plots (**b**, **c**, **e**) shows the median with 25th and 75th percentile borders. Whiskers extend from minimum to maximum. ***p* < 0.001; n.s. not significant (the Mann–Whitney test)

To interrogate why few Merkel cells were formed in *Fgfr2* cKO mice, we analysed embryos at E14.5, the time of Merkel cell specification. By performing immunofluorescence studies, we observed incomplete loss of FGFR2 protein from the skin epithelium and the hair placode cells of E14.5 *Fgfr2* cKO embryos (Supplementary Fig. 4k). This suggests that the residual level of FGFR2 present at the time of Merkel cell specification could allow for a few Merkel cells to form.

**FGFR2 is not required for the specification of HFSCs**. As our data showed that FGFR2 is critical for Merkel cell formation, we next asked if FGFR2 is required for the specification of SOX9(+) cells to HFSCs. Our analysis of primary hair follicle development revealed that while there was a reduction in primary hair follicle length in E16 and P0 *Fgfr2*-null skins compared to controls (Fig. 5a, b and Supplementary Fig. 5a), the number of primary hair follicles was comparable between *Fgfr2* cKO and control skins (Fig. 5c). While *Fgfr2* cKO hair follicles were shorter, their development was largely normal. Indeed, hair follicles in *Fgfr2* cKO skin expressed proper hair follicle differentiation markers AE15, which labels the inner root sheath and medulla[37,38], and AE13, which marks the cuticle and cortex of the hair shaft[37,38] (Supplementary Fig. 5b, c). Furthermore, immunofluorescence analysis also showed that SOX9(+) cells were present in both control and *Fgfr2*-null hair follicles (Fig. 5d–f). Thus, we concluded that the development and cell differentiation were not arrested in *Fgfr2* cKO hair follicles.

In the adult organism, HFSCs promote hair follicle growth through the hair cycle, give rise to sebaceous glands, and express HFSC markers, such as CD34[39]. As newborn *Fgfr2* cKO pups fail to form milk spots and die shortly after birth, in order to analyse HFSCs in the postnatal skin, we performed full-thickness grafts of P0 *Fgfr2* cKO and control skins onto the backs of recipient *Nude* mice[20]. The analysis of grafts 35 days after engraftment (P35) showed that control and *Fgfr2*-null hair follicles were fully developed, and formed hair shafts and sebaceous glands (Fig. 5h±i and Supplementary Fig. 5d, e). Similarly to controls, *Fgfr2*-null hair follicles had a club hair, indicating that these hair follicles had progressed through one hair cycle (Fig. 5i). Furthermore, immunofluorescence staining for HFSC markers CD34 and SOX9 revealed that both control and *Fgfr2*-null hair follicle bulge cells expressed SOX9 (Fig. 5h) and CD34 (Fig. 5i). Together, these analyses show that FGFR2-mediated signalling is not necessary for HFSC specification.

## Discussion

In this paper, we elucidated the cell of origin of Merkel cells and identified the molecular pathways controlling Merkel cell and hair

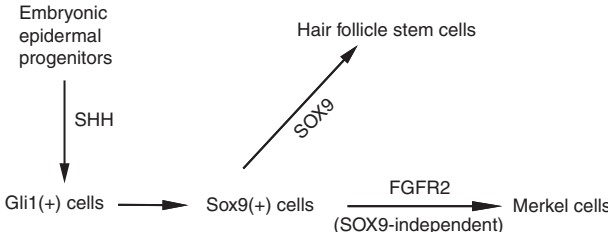

**Fig. 6** Working model of the Merkel cell specification process. In the developing skin, SHH signalling is required to specify SOX9(+) cells in hair placodes. SOX9(+) cells give rise to both HFSCs and Merkel cells. While SOX9 is required for SOX9(+) cell specification to HFSCs, it is dispensable for Merkel cell formation. Conversely, FGFR2-mediated skin epithelial signalling is required for Merkel cell formation but is not required for HFSC specification

follicle lineage formation (Fig. 6). Our data show that Merkel cells originate from cells located inside hair placodes, which is interesting, as ectodermal placode structures are known to give rise to different sensory tissues[40]. For example, taste sensory cells are known to originate from an ectodermal placode[41]. The development of inner ear hair cells that are responsible for balance and the perception of sounds also starts with the formation of an ectodermal placode[42]. Our data now show that similarly to other sensory epithelial tissues, Merkel cells that are involved in light-touch sensations are also formed from ectodermal placodes.

By performing fate-mapping experiments, we show that SOX9 (+) and not LHX2(+) cells preferentially gives rise to Merkel cells. Previous lineage tracing experiments showed that progenies of SHH-expressing cells do not give rise to Merkel cells[24]. As SHH-expressing cells also express LHX2[43], our LHX2 fate-mapping results are consistent with these data. Furthermore, previous studies using *Shh*[GFPcre/+]; *R26*[YFP/+] mice[44] and our studies using *Lhx2-CreER*; *tdTomato* mice have shown to map the majority of cells of the hair follicle lineage. As Merkel cells are largely not labelled using either *Shh-Cre* or *Lhx2-CreER* fate-mapping strategies, these data indicate that Merkel cells originate from outside of the SHH- and LHX2-derived hair follicle lineages, which is consistent with previous findings[24]. Intriguingly, a few Merkel cells appear to be labelled in *Shh-Cre*[24] and *Lhx2-CreER* lineage tracing experiments. This is likely due to recent findings showing that during development some SOX9(+) cells originate from SHH(+) cells[19]. Alternatively, a gradient of expression of SOX9 and LHX2 observed in developing hair follicles leads to a few SOX9 and LHX2 double-positive cells at the transitional zone[22], and might thus result in labelling of Merkel cells in the *Lhx2-CreER* lineage tracing experiment. Taking our study and the published data together, we can conclude that Merkel cell progenitors are predominantly SOX9(+) SHH(−) LHX2(−) cells. While SOX9(+) cells give rise to Merkel cells, our data show that the SOX9 protein itself is absent from ATOH1(+) Merkel cells and SOX9 is not required for Merkel cell specification.

SOX9(+) cells were shown to give rise to developing hair follicles and to serve as precursors of adult HFSCs, which maintain the growth of postnatal follicles during the hair cycle[20]. Our data now show that Sox9(+) cells have an additional function to give rise to Merkel cells. It is intriguing that SOX9(+) cells gives rise to two completely different structures, the hair follicles and Merkel cells, which perform completely different functions in the body. Furthermore, during development, hair follicles grow downward into the underlying dermis, while Merkel cells are located upwards in the basal layer of the epidermis. Despite these differences, genetic studies showed a strong correlation between hair follicle and Merkel cell development, as genetic mutations ablating hair follicle formation also lead to the loss of Merkel cells[23,24]. Our data now provide an explanation for the observed genetic similarities. We show that the specification of SOX9(+) cells is affected in mutants in which hair formation is abrogated, resulting in hair follicle defects and loss of Merkel cells. Future studies of the SOX9(+) cell population will be needed to examine how these cells are fated to become such diverse cell types in the skin. One possible explanation for this is the heterogeneity within the SOX9(+) cell population, as it has recently been shown that some SOX9(+) cells express KRT79, a marker of terminally differentiated hair follicle cells[45,46].

Our data reveal different requirements for SOX9 and FGFR2 to specify HFSCs and Merkel cells from SOX9(+) cells. While SOX9 is essential to specify HFSCs[20], our data show that it is not required for Merkel cell formation. We show that FGFR2-mediated signalling in the skin epithelium is critical for Merkel cell formation, but is not required for HFSC specification. The importance of skin epithelial FGFR2-mediated signalling for

Merkel cell development is very interesting. Among four FGF receptors, only FGFR2 is expressed at high levels in the developing skin epithelium at the time of Merkel cell specification and, importantly, we show that FGFR2 is expressed in SOX9(+) cells and in Merkel cells.

Which FGF ligand controls Merkel cell specification? We show that FGF20 is important for this process, as there is a reduction in the number of Merkel cells in *Fgf20*-null skins. Recently, Xiao and colleagues[24] observed a drastic reduction in the expression of a key Merkel cell differentiation factor, ATOH1, in E15.5 *Fgf20* KO compared to control. Interestingly, however, despite the reduction in ATOH1 expression at the time of Merkel cell specification, Xiao and colleagues[24] observed small reduction in Merkel cell numbers in P0 *Fgf20* KO mice compared to controls. While this result is different to our findings, this discrepancy could be due to differences in mouse genetic backgrounds used in two studies and thus for some mouse genetic backgrounds lacking FGF20, other FGF ligands can sufficiently compensate for the loss of FGF20 and specify Merkel cells. Indeed, there are more than 23 different known genes that code for FGFs, and much redundancy between FGFs has been reported in other developmental systems[47]. While our data show that FGF10 is not required for Merkel cell formation, other FGF ligands might function with FGF20 to play a role in this process.

Overall, our findings uncover a cell of origin of Merkel cells and identify the molecular mechanisms controlling Merkel cell development. Uncovering the biology of Merkel cells is critical for understanding the function of the skin as a sensory organ, and the interaction between the skin and the nervous system[2]. It is also essential for determining the mechanisms of our perception of touch, which is critical for our survival, as well as the biological processes that go awry in children with hypo and hyper tactile sensitivities, including those observed in autism spectrum disorders[3,5,48–50]. Finally, defining the mechanisms controlling Merkel cell formation will also allow us to devise differentiation protocols to produce Merkel cells in vitro, which is still challenging. This will be important for improving upon our current method of in vitro epidermis production, allowing us to generate epidermis that is capable of providing both barrier and sensory functions[51].

## Methods

**Mice.** All mice were housed at the Center for Comparative Medicine and Surgery (CCMS), Icahn School of Medicine at Mount Sinai (ISMMS) and cared for according to the Institutional Animal Care and Use Committee (IACUC)-approved protocol LA11-0020. For research purposes and in cases of veterinarian-monitored illness, we use carbon dioxide in accordance with the Panel on Euthanasia of the American Veterinary Medical Association to euthanize animals. At least three animals from independent litters were used for each analysis. Immunocompromised *Nude* (NU-Foxn1^Nu) mice and wild-type CD1 mice were obtained from Charles River Laboratory. *Fgf20*^β-gal mice[52] were obtained from David M. Ornitz, and were maintained on a mixed genetic background. *Fgf10*-null mice have been described[53]. *Fgfr2*^flox and *Fgf2-mCherry* reporter mice[36] were generously provided by Philippe Soriano. *Sox9-CreER* was described[54]. *Lhx2-CreER* mice were obtained from Josh Huang and will be described elsewhere. *R26-mT/mG* (Stock number: 007676), *Sox9*^flox (Stock number: 013106), *Nfatc1*^flox (Stock number: 022786), *Smo*^flox (Stock number: 004526), *Atoh1-GFP* (Stock number: 013593), *R26-rtTA* (Stock number: 006965), and *Krt14-Cre* (Stock number: 004782) were obtained from The Jackson Laboratory. Because *Fgfr2* conditional KO mice die shortly after birth, all analyses of these mice after P0 were performed on grafted skin obtained using the full-thickness grafting protocols[20,55,56]. Briefly, full-thickness skins from the back of sex-matched wild type and *Fgfr2* conditional knockout (cKO) P0 mice were removed and grafted onto the backs of anaesthetised female *Nude* mice (nu/nu), with each recipient receiving a wild-type and cKO graft. Grafts were secured by sterile gauze and cloth bandages, which were removed after 4 weeks. For Tamoxifen treatment, *Sox9-CreER; R26-mT/mG* and *Lhx2-CreER; R26-tdTomato* were injected with Tamoxifen (Sigma-Aldrich; St. Louis, MO) doses totalling 40 μg per gram body weight at E13.5 and E14.5, and pups were collected at E16 or P0 for further analysis. Mice were genotyped by PCR using DNA extracted from tail skin. Primers used for genotyping are provided in Supplementary Table 1.

**In utero injections.** *R26-rtTA* male mice were mated to CD1 female mice to generate *R26-rtTA* embryos for ultrasound guided lentiviral injection[57]. A high titre (>10^9 CFU) of inducible SHH overexpression lentiviral construct (LV-TRE-SHH-PGK-H2B-RFP)[58] was used to perform microinjections into the amniotic cavities of E9 embryos. E9 timed pregnant mice were anaesthetised with 2.5% isoflurane and 1% oxygen, and then positioned on a mouse platform (Integrated Rail System, VisualSonics). A midline incision was performed, and uterine horns were gently exteriorized through the incision and carefully drawn through a Parafilm™ flap in the bottom of a sterilized Petri dish. Four to five embryos were injected using a micropipette, with 1 μl of SHH overexpression lentivirus for each embryo, and the uterine horn was reinserted into the peritoneal cavity. The abdominal wall and skin were closed with sutures. Expression of the viral construct was induced at E12 by gavage treatment of 200 μl doxycycline (10 mg per ml in sterile water (Sigma-Aldrich) to the mice that were pregnant with the injected pups). The pregnant mice were then fed doxycycline chow (200 mg per kg, Bio-Serv) for 5 days. SHH overexpressed and control embryos were collected at E17 for further analysis.

**Immunofluorescence and microscopy.** For immunofluorescence, tissues were collected from mice, embedded fresh into OCT (Tissue-Tek; Torrance, CA), and subsequently cut into 10 μm sections using a Leica Cryostat. Slides were fixed for 10 min in 4% paraformaldehyde (PFA; Electron Microscopy Sciences) in phosphate-buffered saline (PBS) and blocked for 1 h or overnight in PBS with 1% Triton X-100, 1% bovine serum albumin, and 0.25% normal donkey serum. Embryos collected after lentiviral infection for the SHH overexpression experiment were pre-fixed for 7 min in 4% PFA at room temperature. Primary antibodies were diluted in blocking solution and incubations were carried out for 1 h or overnight, followed by incubation in secondary antibodies for 1 h at room temperature. Slides were then counterstained with DAPI and mounted using antifade mounting media. For whole-mount immunofluorescence, back skins were collected from newborn mice and placed in 1.26 U per mL dispase (Invitrogen; Carlsbad, CA) for 4 h at 4 °C. Then, the epidermis was peeled from the dermis and fixed in 4% PFA for 1 h at 4 °C. Skins were blocked with blocking solution for 2 h at room temperature. Primary antibodies were diluted in blocking solution and incubations were carried out overnight at 4 °C, followed by incubation in secondary antibodies for 2 h at room temperature. Skins were then counterstained with DAPI or Hoechst and mounted in antifade mounting media for imaging. TUNEL stainings were performed using the Roche TUNEL kit (Roche Diagnostics) according to the manufacturer's instructions. Slides were imaged using a Leica DM5500 upright slide microscope or Confocal Zeiss LSM880 Airyscan microscope using ×10, ×20, or ×40 objectives.

**Antibodies.** Antibodies were used as follows: KRT14 (generous gift of Julie Segre, National Human Genome Research Institute, MD, USA, 1/20,000); KRT8 (Developmental Studies Hybridoma Bank, TROMA-1, 1/500); KRT5 and LHX2 (generous gift of Elaine Fuchs, The Rockefeller University, NY, USA, 1/500); KRT20 (Dako, M7019, 1/70); SOX2 (Stemgent, 09–0024, 1/150); GFP (Abcam, ab13970, 1/1000); RFP (MBL, PM005, 1/4000); E-CADHERIN (Invitrogen, 131900, 1/2000); SOX9 (Abcam, ab185966, 1/500); β-GALACTOSIDASE (Abcam, ab9361, 1/500); CD34 (eBioscience, 14-0341-82, 1/250); AE15 (Santa Cruz, sc 80607, 1/100); AE13 (abcam, ab16113, 1/20); FGFR2 (Cell Signaling, 23328, 1/150); P-CAD (Fisher, BAF761, 1/400); EPCAM (BD Bioscience, 552370, 1/500). For IF, secondary antibodies coupled to Alexa 488, 549, or 649 were obtained from Jackson Laboratories (1/1000).

**Quantifications.** For quantification of Merkel cells per mm of skin[59], the length of each section was measured and the number of positively stained cells was counted. Typical section lengths were between 7 and 14 mm. We counted a large number of Merkel cells in the control conditions (>300 KRT8(+) cells) and then counted the number of Merkel cells in a similar length of skin for each mutant line. Typically, at least 100 mm of skin were counted for each condition. Quantification of touch domes (TD): skin samples were stained with KRT8 and six non-overlapping images of 6.55 (Field of view or 2.56 × 2.56) mm² areas of P0 skin per animal were quantified. The mean was calculated and used in statistical analysis. Number of animals used, n≥4. For quantification of the number of Merkel cells per TD, 7 to 10 TDs were randomly chosen from each skin sample. The length of the hair follicles was measured by drawing a line that starts from the bottom of the basal layer and goes to the end of the hair follicle, and measuring the length of the line. Hair follicle length is presented in μm.

**Statistics.** In all column bar graphs, mean value±one standard deviation is presented. Box-and-whisker plots show first to third quartiles around the median, with whiskers showing minimum–maximum range and outliers presented as individual data points. The number of biological replicates used for comparison is indicated in each figure. To determine the significance between two groups in all experiments, the Mann–Whitney test was performed (GraphPad Prism 5). For all statistical tests, the $p < 0.05$ level of confidence was accepted for statistical significance, and actual $p$-values (to four decimal places) were provided in the figure legends. Significance levels were defined as *$p < 0.05$; **$p < 0.01$; ***$p < 0.001$; n.s., not significant.

**Data Availability**. The data that support the findings of this study are available from the corresponding author upon reasonable request.

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

## Acknowledgements

For help, critical suggestions, reagents, and experimental input, we are grateful to Drs. Elaine Fuchs, Julie Segre, Michael Rendl, Chenleng Cai, Rui Yi, Sergei Ezhkov, and Carolina Perdigoto. We would like to thank Dr. Philippe Soriano for his generous gift of the *Fgfr2* floxed and *Fgfr2-mCherry* mice. We also thank the personnel of the Microscope Facility at the Icahn School of Medicine at Mount Sinai and Light Microscopy Unit of the Institute of Biotechnology, University of Helsinki. C.B. is a Merksammer fund scholar. K. L.D.-D. was a trainee of the NIDCR-Interdisciplinary Training Program in Systems and Developmental Biology and Birth Defects T32HD075735. Research reported in this publication was supported by the National Institute of Arthritis and Musculoskeletal and Skin Diseases of the National Institutes of Health under award numbers R01 AR063724 (to E.E.) and in part by R01 AR070825 (to Y.-C.H.). This work was funded in part by NIH R35-DE026602 (to O.D.K.) and Sigrid Juselius Foundation and Academy of Finland (Grant no. 1272280) (to M.L.M.). We also thank for NCI P30 Cancer Center Support Grant (CCSG). The content is solely the responsibility of the authors and does not necessarily represent the official views of the National Institutes of Health. This work was funded in part by the Smith Family Awards Program, Basil O'Connor Starter Scholar Research Awards, and the Pew Scholar (to Y.-C.H.).

## Author contributions

M.B.N., I.C., and E.E. conceived the experimental plan. M.B.N., I.C., V.K., Z.X., C.B., K.L. D-D., P.M., P-C.T., O.D.K., Y-C.H., M.L.M., and E.E. carried out the experiments. M.B. N., I.C., and E.E. wrote the manuscript with input from all authors.

## Additional information

**Competing interests:** The authors declare no competing interests.

