## [Peer Review File · Nature Communications]

Reviewers' comments:

Reviewer #1 (Remarks to the Author):

In this manuscript, Nguyen and colleagues have identified a key regulator of Merkel cell development. This study combines lineage tracing, knockout models and excellent histology to map out pathways that are critical for Merkel cell formation. The exact origin of these specialized skin sensory cells has long been debated, and the field is slowly filling in the mechanistic puzzle pieces. This manuscript represents an important step forward in understanding this process, and the findings are sound, but it would fit best in a more specialized journal.

The finding that *Fgfr2* signaling distinguishes HFSC from Merkel cell lineage in the *Sox9+* population is remarkable, and perhaps the first pathway to be defined that so clearly separates these fates. It is also an important extension of previous work showing that *Shh* is important for Merkel cell specification.

Nonetheless, the involvement of *Shh*, as well as *Fgf20* signaling in this process had been previously studied. Moreover, the deletion of functional *Fgfr2* results in an impressive, but not a complete loss of Merkel cells. Many Merkel cells are still present according to the quantification and the whole-mount staining. There is not a discussion surrounding why this might be. Although other ligands can compensate for loss of *Fgf20*, the authors suggest that there are no other *Fgf* receptors in the developing skin. Is there another pathway that compensates or is this due to incomplete recombination? Is it possible that Merkel cell specification is simply delayed because the hair follicles are growing at a slower rate? To exclude this possibility, it would be interesting to see whether skin that is analyzed at P35 in the grafting experiments has recovered Merkel cell numbers. Overexpression or ectopic expression of *Fgfr2* would also test sufficiency.

Interestingly, the results presented here for disrupted *Fgf20* signaling are different than previously published (Xiao 2016). The difference in these findings is not discussed. Moreover, the authors also make opposing conclusions from this paper regarding the origin of Merkel cell lineage in the placode, where previous study concluded it was derived outside of the placode. Again, the differences in these conclusions are not discussed.

The data in this paper are sound and will represent an important addition to the field, but to accommodate further comparisons and in order to appropriately put these data in context, we feel that a journal that allows for longer format and more discussion would better suit this manuscript.

Reviewer #2 (Remarks to the Author):

The authors report on developmental studies of sensory Merkel cells in mouse skin. They focus on *Sox9*, a transcription factor expressed in developing hair follicles and required for normal hair development. They find that *Sox9* cells in the embryonic epidermis give rise to Merkel cells and this requires *Fgfr2* signaling. At the same time, they show that the *Lhx2* progenitors also found in

developing hair follicles do not generally make Merkel cells, that Sox9 function is not necessary for Merkel cell specification, and that Fgfr2 signaling is dispensable in hair follicle formation. Moreover, Shh is implicated as an upstream regulator of Sox9 in these developmental processes. This study is an elegant in situ dissection of the cells and signaling pathways involved in Merkel cell specification, and it fills in critical features in our understanding of the complex reciprocal processes that control mammalian tissue specification during development. Understanding the fundamentals of developmental lineage specification has implications in unraveling disease states, developing stem cell based therapies, and identifying therapeutic targets for differentiated cancers – in this case, Merkel cell carcinoma.

Major Comment:

1. The interpretation of these results depends partly on semantics and how one defines tissue domains/structures. One can use traditional anatomical definitions for structures, but in modern times we tend to move towards defining structures by molecular signatures and functional distinctions. In the skin epithelium, there are multiple lineages including the cycling hair follicle, the isthmus of the hair follicle, the interfollicular epidermis, and the touch dome with its Merkel cells. Each of these lineages has their own molecular markers and are maintained by their own resident stem cells. Merkel cells were originally posited to arise in hair follicles based on K8 staining within hair germs of developing primary hair follicles [Acta Anat (Basel). 1995;152(2):93-109.]. Later it was discovered that, although the Merkel cells arose in these anatomical structures, they formed in a molecularly distinct compartment (NCAM+) on the caudal side of forming follicles. This region was primarily derived from cells outside of the hair placode lineage that goes on to form the mature hair follicle (Shh-Cre constitutive fate map lineage)[PLoS Genet. 2016 Jul 14;12(7):e1006150.]. This is consistent with what was observed in the Lhx2-CreER fate map – although the authors did not fully characterize the follicular contributions of this embryonic lineage, it appeared that the majority of the hair follicle was labeled. The authors convincingly showed that Lhx2-CreER fate-mapped cells infrequently give rise to Merkel cells. The Shh and Lhx2 results both suggest that Merkel cells are largely specified from outside the hair follicle lineage, but some Merkel cell progenitors do express these hair placode markers. The present study provides evidence that the touch dome anlage is indeed adjacent to the hair follicle anlage, and it is comprised largely of Sox9+, Lhx2-, Shh- cells. Thus, Sox9 expressing cells in the E14-E15 epidermis can contribute to both the hair follicle and the touch dome/Merkel cells. However, taken together, it seems that Merkel cells are generally not arising from the *same* Sox9 cells that give rise to the follicle. The authors may want to consider this interpretation of the data and refrain from concluding that Merkel cells arise from Sox9+ hair follicle/placode progenitors because, while this is true anatomically, it ignores the functional lineage data that suggest the touch dome and primary hair follicle arise from distinct, adjacent embryonic lineages that slightly overlap. A model where there are molecular, lineage, and fate differences between adjacent touch dome and primary hair follicle progenitor populations is also consistent with their differential dependence on Fgfr2 signaling.

Minor comments:

2. Introduction – “The skin epithelium consists of the epidermis, ..., hair follicles, ..., and Merkel cells ...” is too much of an oversimplification that omits melanocytes, Langerhans cells, nerve endings,

eccrine, apocrine, sebaceous glands, etc.

3. Introduction and results – “surprisingly” and “unexpectedly” finding Merkel cells associated with developing hair follicles seems to ignore prior reports of the same. [Acta Anat (Basel). 1995;152(2):93-109.]

4. Results – “Lhx2(+) cells, which will give rise to the terminally differentiated inner layers of hair follicles”. The cited references do not appear to include any fate mapping results to support this statement. Moreover, the statement seems to contradict the follicle labeling in Figure 2d.

5. Results, Figure 1c,d – please specify if any of the Atoh1-GFP cells co-labeled with Sox9.

6. Results, Figure S1 – the text implies that the Lhx2 fate map was also analyzed at E16, but it seems only the Sox9 fate map was collected at that time point.

7. Results – “Consistent with previous findings,” cites reference 19, but reference 27 is more relevant to the statement.

8. Results – amniotic fluid is more commonly used than “amniotic liquids”.

9. Results, Figure S2c – Were the ectopic Merkel cells also RFP and Sox9 positive? Wouldn't that be different from native Merkel cells?

10. Results – “Shh signalling promotes Sox9(+) cell specification and thus controls Merkel cell formation.” Since you found that Sox9 was dispensable in Merkel cell formation, using “and thus” does not seem appropriate.

11. Results – please indicate the alleles used to make the Nfatc1 cKO mouse.

12. Results – Fgf20 “is required for primary and secondary hair follicle down-growth”. In the cited reference primary and secondary hair shaft defects were measured, and a defect in primary follicle down-growth was measured. Where was a defect in secondary down-growth demonstrated?

13. Results, Figure 4d-f – The magnitude of the Merkel cell defect in the FGF20 null mouse is different from what was reported in reference 23. How do you account for this?

14. Results – “Sox2 staining was only observed in the dermal condensates”. This seems to contradict the small number of Sox2 cells in the epidermis reported in Figure 4i.

15. Results – “The reduced number of Merkel cells was not due to apoptotic cell death, as assessed by TUNEL staining”. This statement is too definitive for a single time point assessment. Please make the statement more accurate.

16. Results, Figure 5b – please define how follicle length was measured. Was epidermal thickness included, and how were differences in epidermal thickness accounted for?

17. References – citation 21 is actually: Development. 2011 Nov;138(22):4843-52

Reviewer #3 (Remarks to the Author):

The authors found that Merkel cells originate from Sox9+ hair placode cells by lineage tracing. Unlike HFSC, Sox9 expression is not required for Merkel cell formation. They also found that Shh signaling is essential for Merkel cell formation by the specification of Sox9+ cells. In addition, Fgfr2 signaling is required for Merkel cell formation but not for HFSC formation. In sum, Shh signaling specifies Sox9+ cells in placode and some of Sox9+ placode cells become Merkel cells through Fgf signaling. The manuscript is well organized to demonstrate these important findings effectively. The manuscript will significantly advance our understanding of Merkel cell development and I would like to recommend this to be published in this journal. Addressing the following points and questions will further clarify the model and improve the manuscript.

1. The authors show that both Sox9 negative cells and Lhx2 positive cells can give rise to Merkel cells in Fig. 2c and e. This needs some clarifications. First, regarding Sox9 negative cells, is it because of incomplete recombination or other epithelial cells? Second, do Lhx2 positive cells express Sox9 as well at earlier time point?
2. In Fig. 1c and d, Atoh1+Merkel cells lack Sox9 at E14.5 and E15.5. Nonetheless, the lineage tracing of Sox9 cells shows Sox9+ cells give rise to Merkel cells. It suggests that Sox9 expression in Merkel cell progenitor is very transient and the authors can analyze or mention this.
3. Sox9+ cells are absent in Smo-null skin epithelium at P0. Can the authors confirm the Sox9 expression at earlier than P0 (e.g. E13.5) given Sox9 expression may be transient in wild type skin?
4. Is Fgfr2 expression confined to Merkel cell progenitor?
5. Is Sox9+ cells detected in Fgf20-null skin like Fgfr2-null skin? In addition, it would be more convincing if quantifications of Sox9+ cells in Fgf20- and Fgfr2-null skin are added.
6. Does Fgf20 act in autocrine manner in Fgfr2+ cells? or do other cells type secrete Fgf20?
7. Does Sox9 repress Fgf20 or in the cells giving rise to Merkel cells?
8. Labels are missing; Fig. 4g and Fig. 5h

Reviewer #1 (Remarks to the Author):

In this manuscript, Nguyen and colleagues have identified a key regulator of Merkel cell development. This study combines lineage tracing, knockout models and excellent histology to map out pathways that are critical for Merkel cell formation. The exact origin of these specialized skin sensory cells has long been debated, and the field is slowly filling in the mechanistic puzzle pieces. This manuscript represents an important step forward in understanding this process, and the findings are sound, but it would fit best in a more specialized journal. The finding that Fgfr2 signaling distinguishes HFSC from Merkel cell lineage in the Sox9+ population is remarkable, and perhaps the first pathway to be defined that so clearly separates these fates. It is also an important extension of previous work showing that Shh is important for Merkel cell specification.

We are grateful to reviewer #1 for recognizing the importance of our work showing the origin of Merkel cells.

Nonetheless, the involvement of Shh, as well as smoothed and Fgf20 signaling in this process had been previously studied. Moreover, the deletion of functional Fgfr2 results in an impressive, but not a complete loss of Merkel cells. Many Merkel cells are still present according to the quantification and the whole-mount staining. There is not a discussion surrounding why this might be. Although other ligands can compensate for loss of Fgf20, the authors suggest that there are no other Fgf receptors in the developing skin. Is there another pathway that compensates or is this due to incomplete recombination?

We are thankful to reviewer #1 for this comment. We performed immunofluorescence analysis of Fgfr2 in control and Fgf2 cKO embryos to determine whether Fgfr2 is efficiently ablated from the skin epithelium. We could not detect any Fgfr2 in E16 Fgfr2cKO embryos, suggesting that by E16, Fgfr2 is efficiently ablated from the Fgfr2cKO skin epithelium (Supplementary Figure 4j). In contrast, immunofluorescence analysis of E14.5 embryos showed residual level of Fgfr2 in Fgfr2 cKO embryos compared to controls (Supplementary Figure 4k). These data indicate that at E14, when Merkel cell specification occurs, Fgfr2 is not completely lost from the skin epithelium of Fgfr2 cKO embryos, and this can explain how the specification of some Merkel cells occurs. We included these data in the revised manuscript and the text now reads as follows: **“To interrogate why few Merkel cells were formed in Fgfr2 cKO mice, we analysed embryos at E14.5, the time of Merkel cell specification. By performing immunofluorescence studies, we observed incomplete loss of Fgfr2 protein from the skin epithelium and the hair placode cells of E14.5 Fgfr2 cKO embryos (Supplementary Fig. 4k). This suggests that the residual level of Fgfr2 present at the time of Merkel cell specification could allow for a few Merkel cells to form.”**

Is it possible that Merkel cell specification is simply delayed because the hair follicles are growing at a slower rate? To exclude this possibility, it would be interesting to see whether skin that is analyzed at P35 in the grafting experiments has recovered Merkel cell numbers. Overexpression or ectopic expression of Fgfr2 would also test sufficiency.

Our previous studies showed that the Merkel cells are depleted in grafted control mouse skins (PMID:23673358; Figure 1a, b). In that paper, to analyse Ezh1/2 2KO skins postnatally, we had to perform skin grafts. As you can see from Figure 1a and 1b of this paper, the number of Merkel cells in the P90 skin is practically undetectable. This is likely due to the fact that the maintenance of Merkel cells in postnatal mice is

dependent on Sonic hedgehog (Shh) supplied by innervating nerves (PMID:26015562), and skins are denervated during the engraftment procedure. While we performed immunofluorescence staining for Merkel cell markers on P35 engrafted skins, we could not detect any Merkel cells, even in control skins. We thus could not assess whether there is a difference in the number of Merkel cells between control and Fgfr2 cKO at P35.

While it is certainly an interesting question if Fgfr2 is sufficient to specify Merkel cells, answering this question is outside of the scope of this study, as it would require the generation of a transgenic mouse line.

Finally, because Fgfr2 cKO hair follicles were shorter, we checked if there is a delay in their development. While P0 Fgfr2 cKO hair follicles were shorter, they did express proper hair follicle differentiation markers AE15, which labels the IRS and medulla, and AE13, which marks the cuticle and cortex of the hair shaft (Supplementary Fig. 5b, c). Importantly, these markers of hair follicle differentiation are absent at the time point when Merkel cells are specified and appear later in development (PMID: 14610062 and PMID: 28670819). Thus, we concluded that hair follicle development growth rate in Fgfr2cKO is not slower compared to the control rate. We included these data in the revised manuscript and the text now reads as follows: “While Fgfr2 cKO hair follicles were shorter, their development was largely normal. Indeed, hair follicles in Fgfr2 cKO skin expressed proper hair follicle differentiation markers AE15, which labels the inner root sheath (IRS) and medulla^{36,37}, and AE13, which marks the cuticle and cortex of the hair shaft^{36,37} (Supplementary Fig. 5 b, c).”

Interestingly, the results presented here for disrupted Fgf20 signaling are different than previously published (Xiao 2016). The difference in these findings is not discussed.

We agree with reviewer #1 that our analysis of Fgf20 KO mice showed different results compared to those previously published by Xiao 2016. It is fair to say that we do not really know the reason for the discrepancy. We suspect that difference in the background of the Fgf20-null mice used could be one such reason, given that background effect is a relatively common phenotype in mouse knockouts (PMID: 19266333). We have updated the information about the background of the Fgf20 KO mice used in our studies in Materials and Methods. We also included the following text in the discussion section of our revised manuscript. “Recently, Xiao and colleagues²³ observed a drastic reduction in the expression of a key Merkel cell differentiation factor, Atoh1, in E15.5 Fgf20 KO compared to control. Interestingly, however, despite the reduction in Atoh1 expression at the time of Merkel cell specification, Xiao and colleagues²³ observed no changes in Merkel cell numbers in P0 Fgf20 KO mice compared to controls. While this result is different to our findings, this discrepancy could be due to differences in mouse genetic backgrounds used in two studies and thus, for some mouse genetic backgrounds lacking Fgf20, other Fgf ligands can sufficiently compensate for the loss of Fgf20 and specify Merkel cells.”

Moreover, the authors also make opposing conclusions from this paper regarding the origin of Merkel cell lineage in the placode, where previous study concluded it was derived outside of the placode. Again, the differences in these conclusions are not discussed.

We are grateful to reviewer #1 for bringing up this important point. We do not think that our data oppose to conclusions in the Xiao 2016 publication. We think the differences are largely due to semantics and how one discriminates between anatomical structure vs. lineage cell fate. Our data on Lhx2 lineage tracing are, in fact, consistent with the data from Xiao 2016, in which Shh-Cre lineage tracing showed that while Shh(+) and Lhx2(+) give rise to the majority of cells within hair follicles, Shh(+) and Lhx2(+) have minimum contribution to Merkel cells. Thus, it is fair to conclude that Merkel cells largely originate outside of the Shh- and Lhx2-derived hair follicle lineages. We show that Merkel cells appear inside of developing hair follicle anatomical structures, and Sox9(+) cells give rise to majority of Merkel cells. Considering the data from Xiao 2016 and our data together, we can conclude that Merkel cells largely originate from Lhx2(-) Shh(-) Sox9(+) cells. We have added the following text to the discussion section of our revised manuscript: “Furthermore, previous studies using Shh^{GFPcre/+}; R26^{YFP/+} mice⁴³ and our studies using Lhx2-CreER; tdTomato mice have shown to map the majority of cells of the hair follicle lineage. As Merkel cells are largely not labelled using either Shh-Cre or Lhx2-CreER fate mapping strategies, these data indicate that Merkel cells originate from outside of the Shh- and Lhx2-derived hair follicle lineages, which is consistent with previous findings²³ ... Taking our study and the published data together, we can conclude that Merkel cell progenitors are Sox9(+) Shh(-) Lhx2(-) cells.”

The data in this paper are sound and will represent an important addition to the field, but to accommodate further comparisons and in order to appropriately put these data in context, we feel that a journal that allows for longer format and more discussion would better suit this manuscript.

We are thankful to reviewer #1 for recognizing the importance of our paper for the field. To put our data in context, we have expanded the discussion section of our revised paper.

Reviewer #2 (Remarks to the Author):

The authors report on developmental studies of sensory Merkel cells in mouse skin. They focus on Sox9, a transcription factor expressed in developing hair follicles and required for normal hair development. They find that Sox9 cells in the embryonic epidermis give rise to Merkel cells and this requires Fgfr2 signaling. At the same time, they show that the Lhx2 progenitors also found in developing hair follicles do not generally make Merkel cells, that Sox9 function is not necessary for Merkel cell specification, and that Fgfr2 signaling is dispensable in hair follicle formation. Moreover, Shh is implicated as an upstream regulator of Sox9 in these developmental processes. This study is an elegant in situ dissection of the cells and signaling pathways involved in Merkel cell specification, and it fills in critical features in our understanding of the complex reciprocal processes that control mammalian tissue specification during development. Understanding the fundamentals of developmental lineage specification has implications in unraveling disease states, developing stem cell based therapies, and identifying therapeutic targets for differentiated cancers – in this case, Merkel cell carcinoma.

We are very happy that reviewer #2 recognizes the importance of our work for understanding mammalian skin development, lineage specification, and the possible origin of Merkel cell carcinoma.

Major Comment:

1. *The interpretation of these results depends partly on semantics and how one defines tissue domains/structures. One can use traditional anatomical definitions for structures, but in modern times we tend to move towards defining structures by molecular signatures and functional distinctions. In the skin epithelium, there are multiple lineages including the cycling hair follicle, the isthmus of the hair follicle, the interfollicular epidermis, and the touch dome with its Merkel cells. Each of these lineages has their own molecular markers and are maintained by their own resident stem cells. Merkel cells were originally posited to arise in hair follicles based on K8 staining within hair germs of developing primary hair follicles [Acta Anat (Basel). 1995;152(2):93-109.]. Later it was discovered that, although the Merkel cells arose in these anatomical structures, they formed in a molecularly distinct compartment (NCAM+) on the caudal side of forming follicles. This region was primarily derived from cells outside of the hair placode lineage that goes on to form the mature hair follicle (Shh-Cre constitutive fate map lineage) [PLoS Genet. 2016 Jul 14;12(7):e1006150.]. This is consistent with what was observed in the Lhx2-CreER fate map – although the authors did not fully characterize the follicular contributions of this embryonic lineage, it appeared that the majority of the hair follicle was labeled. The authors convincingly showed that Lhx2-CreER fate-mapped cells infrequently give rise to Merkel cells. The Shh and Lhx2 results both suggest that Merkel cells are largely specified from outside the hair follicle lineage, but some Merkel cell progenitors do express these hair placode markers. The present study provides evidence that the touch dome anlage is indeed adjacent to the hair follicle anlage, and it is comprised largely of Sox9+, Lhx2-, Shh- cells. Thus, Sox9 expressing cells in the E14-E15 epidermis can contribute to both the hair follicle and the touch dome/Merkel cells. However, taken together, it seems that Merkel cells are generally not arising from the *same* Sox9 cells that give rise to the follicle. The authors may want to consider this interpretation of the data and refrain from concluding that Merkel cells arise from Sox9+ hair follicle/placode progenitors because, while this is true anatomically, it ignores the functional lineage data that suggest the touch dome and primary hair follicle arise from distinct, adjacent embryonic lineages that slightly overlap. A model where there are molecular, lineage, and fate differences between adjacent touch dome and primary hair follicle progenitor populations is also consistent with their differential dependence on Fgfr2 signaling.*

We are grateful to reviewer #2 for providing these insightful comments. We agree that based on our presented data, one can not conclude that the “same” Sox9(+) cells that give rise to the hair follicle lineage also give rise to Merkel cells. The Sox9+ cell population might very well be a heterogeneous population of cells that gives rise to different cell types. Some evidence for this has recently been reported by Sunny Wong’s lab in the Cell Rep. 2017 Apr 25;19(4):809-821 paper, in which the authors showed that a subset of Sox9+ cells expresses Krt79 and consists of terminally differentiated hair follicle cells. We added the following text to the discussion section of our revised manuscript: “Future studies of the Sox9(+) cell population will be needed to examine how these cells are fated to become such diverse cell types in the skin. One possible explanation for this is the heterogeneity within the Sox9(+) cell population, as it has recently been shown that some Sox9(+) cells express Krt79, a marker of terminally differentiated hair follicle cells^{44,45}.”

We also included the Acta Anat (Basel). 1995;152(2):93-109 reference in the text. Finally, we expanded our discussion section to put our data in the context of the PLoS Genet. 2016 Jul 14;12(7):e1006150 paper. We agree with reviewer #2 that our study, together with the PLoS Genet. 2016 paper, provides compelling evidence that a population of Sox9(+) Lhx2(-) Shh(-) cells are Merkel cell progenitors. We included the following text in the discussion section of our revised manuscript: “Furthermore, previous studies using Shh^{GFPcre/+}; R26^{YFP/+} mice⁴³ and our studies using Lhx2-CreER; tdTomato mice have shown to map the majority of cells of the hair follicle lineage. As Merkel cells are largely not labelled using either Shh-Cre or Lhx2-CreER fate mapping strategies, these data indicate that Merkel cells originate from outside of the Shh- and Lhx2-derived hair follicle lineages, which is consistent with previous findings²³. Intriguingly, a few Merkel cells appear to be labelled in Shh-Cre²³ and Lhx2-CreER lineage tracing experiments. This is likely due to recent findings showing that during development some Sox9(+) cells originate from Shh(+) cells¹⁹. Alternatively, a gradient of expression of Sox9 and Lhx2 observed in developing hair follicles leads to a few Sox9 and Lhx2 double positive cells at the transitional zone²², and might thus result in labeling of Merkel cells in the Lhx2-CreER lineage tracing experiment. Taking our study and the published data together, we can conclude that Merkel cell progenitors are Sox9(+) Shh(-) Lhx2(-) cells”

Minor comments:

2. Introduction – “The skin epithelium consists of the epidermis, ..., hair follicles, ..., and Merkel cells ...” is too much of an oversimplification that omits melanocytes, Langerhans cells, nerve endings, eccrine, apocrine, sebaceous glands, etc.

Thank you for this comment! We corrected the text to read as follows: “The skin epithelium is an essential barrier that protects the body from the environment, helps to maintain temperature and keep water within the body, and performs sensory functions¹. These activities are largely provided by the epidermis, hair follicles, and Merkel cells, which serve protective barrier functions, provide thermoprotection, and are involved in mechanosensation, respectively^{1,2}”

3. Introduction and results – “surprisingly” and “unexpectedly” finding Merkel cells associated with developing hair follicles seems to ignore prior reports of the same. [Acta Anat (Basel). 1995;152(2):93-109.]

We modified the text, removed the words “surprisingly and “unexpectedly”, and included the [Acta Anat (Basel). 1995;152(2):93-109.] reference. The text now reads as follows: “These data are consistent with previous studies of Krt8, an early Merkel cell marker which appears after Atoh1 induction, showing that Krt8(+) cells are present inside of developing hair follicles at E15¹⁸.”

4. Results – “Lhx2(+) cells, which will give rise to the terminally differentiated inner layers of hair follicles”. The cited references do not appear to include any fate mapping results to support this statement. Moreover, the statement seems to contradict the follicle labeling in Figure 2d.

We are grateful to reviewer #2 for pointing this out. We removed the cited references and rewrote that paragraph. It now reads as: “At the time when the first Merkel cells appear, hair follicle lineage specification has already occurred. The hair follicles contain discrete cell populations with different locations within hair placodes¹⁹. Cells at the leading edge of developing hair follicles express the transcription factor Lhx2, whereas suprabasal hair placode cells are positive for the stem cell pioneer factor Sox9 (Fig. 1b-g)^{20,21}”

5. Results, Figure 1c,d – please specify if any of the *Atoh1*-GFP cells co-labeled with Sox9.

We analysed if *Atoh1*-GFP cells co-label with SOX9 and could not detect any double positive cells. We included images to illustrate this point in Supplementary Fig. 1a-c and included the following sentence in the result section: “Regardless of cell proximity, *Atoh1*-GFP(+) Merkel cells did not co-label with Sox9 (Supplementary Fig. 1a-c) or *Lhx2* (Fig. 1f, g).” We also added the following to the discussion section: “While Sox9(+) cells give rise to Merkel cells, our data show that the Sox9 protein itself is absent from *Atoh1*(+) Merkel cells and Sox9 is not required for Merkel cell specification.”

6. Results, Figure S1 – the text implies that the *Lhx2* fate map was also analyzed at E16, but it seems only the Sox9 fate map was collected at that time point.

We apologize for this mistake. We corrected the text and it now reads as follows: “Embryos were treated with Tamoxifen at E13.5-E14.5, at which time Sox9(+) and *Lhx2*(+) cells are present but Merkel cells have not yet been specified, and collected from both lines at postnatal day (P) 0 (Fig. 2a) and at E16 for Sox9-CreER; R26-mT/mG (Supplementary Fig. 1d).”

7. Results – “Consistent with previous findings,” cites reference 19, but reference 27 is more relevant to the statement.

We corrected the references in the text as suggested.

8. Results – amniotic fluid is more commonly used than “amniotic liquids”.

We corrected this in the text.

9. Results, Figure S2c – Were the ectopic Merkel cells also RFP and Sox9 positive? Wouldn't that be different from native Merkel cells?

We observed that some ectopic Merkel cells are RFP positive, while others are negative (Supplementary Figure 2c’). Similar conclusions were reached for Sox9 expression in the ectopic Merkel cells (Supplementary Figure 2c’). It is important to note that *Shh* overexpression results in hyperproliferation of the epidermis and disorganization of the skin (PMID: 27414999), which could result in the differences observed in ectopic Merkel cells.

10. Results – “*Shh* signalling promotes Sox9(+) cell specification and thus controls Merkel cell formation.” Since you found that Sox9 was dispensable in Merkel cell formation, using “and thus” does not seem appropriate.

We rewrote this sentence, which now reads as: “Together, these results showed that in the skin epithelium, *Shh* signalling promotes specification of Sox9(+) cells and Merkel cell formation.”

11. Results – please indicate the alleles used to make the *Nfatc1* cKO mouse.

We included this information in the text. The sentence now reads as: “To test if *Nfatc1* is required for Merkel cell specification, we generated and analysed skin epithelium-conditional knockout mice of *Nfatc1* (*Nfatc1* cKO) by crossing *Nfatc1* floxed mice with *Krt14*-Cre mice.” These *Nfatc1* floxed mice were obtained from the Jackson Laboratory and contain *loxP* sites flanking exon 3 of the *Nfatc1* gene. We have included the *Nfatc1* floxed mouse stock number information in the Materials and Methods section.

12. Results – *Fgf20* “is required for primary and secondary hair follicle down-growth”. In the cited reference primary and secondary hair shaft defects were measured, and a defect in primary follicle down-growth was measured. Where was a defect in secondary down-growth demonstrated?

In Figure 2C of the cited paper (Huh et al 2013), the authors showed quantifications of different hair types in 3-week-old control and *Fgf20* KO mice. They observed an absence of guard hairs and a decrease in the number

of auchene and awl hairs in Fgf20 KO mice compared to controls. We agree that down-growth is not the correct term here to describe the phenotype. We re-wrote the sentence, which now reads as: “**Fibroblast growth factor (Fgf) 20 is expressed by hair placodes (Supplementary Fig. 4a, b) and is required for primary and secondary hair follicle formation** ³⁰.”

13. *Results, Figure 4d-f – The magnitude of the Merkel cell defect in the FGF20 null mouse is different from what was reported in reference 23. How do you account for this?*

Thank you for pointing this out! It is fair to say that we do not really know the reason for the discrepancy. We suspect that the difference in the background of Fgf20-null mice used could be one such reason, given that background effect is a relatively common phenotype in mouse knockouts (PMID: 19266333). We have updated the information about the background of the Fgf20 KO mice used in our studies in the Materials and Methods. We also included the following text in the discussion section of our revised manuscript: “**Recently, Xiao and colleagues ²³ observed a drastic reduction in the expression of a key Merkel cell differentiation factor, Atoh1, in E15.5 Fgf20 KO compared to control. Interestingly, however, despite the reduction in Atoh1 expression at the time of Merkel cell specification, Xiao and colleagues ²³ observed no changes in Merkel cell numbers in P0 Fgf20 KO mice compared to controls. While this result is different to our findings, this discrepancy could be due to differences in mouse genetic backgrounds used in two studies and thus, for some mouse genetic backgrounds lacking Fgf20, other Fgf ligands can sufficiently compensate for the loss of Fgf20 and specify Merkel cells.**”

14. *Results – “Sox2 staining was only observed in the dermal condensates”. This seems to contradict the small number of Sox2 cells in the epidermis reported in Figure 4i.*

This was a typo, and thank you for pointing it out! We re-wrote the sentence which now reads as: “**Sox2 staining was observed in the dermal condensates of control and Fgfr2-null hair follicles, indicating that epithelial Fgfr2 was not essential for dermal condensate formation (Fig. 4g).**”

15. *Results – “The reduced number of Merkel cells was not due to apoptotic cell death, as assessed by TUNEL staining”. This is statement is too definitive for a single time point assessment. Please make the statement more accurate.*

We corrected this sentence. It now reads as “**The reduced number of Merkel cells observed in E16 Fgfr2 cKO skins was not due to apoptotic cell death, as assessed by TUNEL staining (Supplementary Fig. 4I).**”

16. *Results, Figure 5b – please define how follicle length was measured. Was epidermal thickness included, and how were differences in epidermal thickness accounted for?*

We did not include the epidermis in our quantifications of hair follicle length. The length of the hair follicles was measured by drawing a line that starts from the bottom of the basal layer and goes to the end of the hair follicle, and measuring the length of the line. Hair follicle length is presented in μm . This information is now included in the Materials and Methods/Quantifications section of the revised paper.

17. *References – citation 21 is actually: Development. 2011 Nov;138(22):4843-52*

Thank you for pointing this out! We corrected this citation.

Reviewer #3 (Remarks to the Author):

The authors found that Merkel cells originate from Sox9+ hair placode cells by lineage tracing. Unlike HFSC, Sox9 expression is not required for Merkel cell formation. They also found that Shh signaling is essential for Merkel cell formation by the specification of Sox9+cells. In addition, Fgfr2 signaling is required for Merkel cell formation but not for HFSC formation. In sum, Shh signaling specifies Sox9+cells in placode and some of Sox9+placode cells become Merkel cells through Fgf signaling. The manuscript is well organized to demonstrate these important findings effectively. The manuscript will significantly advance our understanding

of merkel cell development and I would like to recommend this to be published in this journal. Addressing the following points and questions will further clarify the model and improve the manuscript.

We are very grateful to reviewer #3 for recognizing the significance of our work and for recommending that our paper be in Nature Communications.

1. The authors show that both Sox9 negative cells and Lhx2 positive cells can give rise to Merkel cells in Fig. 2c and e. This needs some clarifications. First, regarding Sox9 negative cells, is it because of incomplete recombination or other epithelial cells?

Thank you for raising this question. We performed immunofluorescence analysis of Sox9 in E16 Sox9-CreER; mT/mG mice and observed that around 8% of Sox9(+) cells were GFP-negative, suggesting incomplete recombination using our labelling strategy. We have included these data in the paper and the text now reads as follows: "While we observed that roughly 10% of Krt8(+) cells were not GFP-labelled (Fig. 2b, c and Supplementary Fig. 1d, e), this is likely due to incomplete recombination, as a similar percentage of Sox9(+) cells remained GFP-negative (Supplementary Fig. 1f, g)."

Second, do Lhx2 positive cells express Sox9 as well at earlier time point?

This indeed is an interesting question. Due to the importance of Lhx2 and Sox9 for hair follicle morphogenesis, numerous studies have examined the expression of these transcription factors in developing hair follicles. While it is clear that Sox9 and Lhx2 are expressed in different populations (PMID:26771489 Figure 2F; PMID:18593557 Figure 2B; PMID:22028024 Figure 1C), a gradient of expression of these genes is observed with some cells having intermediate levels of both Lhx2 and Sox9 proteins (PMID:22028024 Figure 1C). We included the following sentence in our revised manuscript: "Alternatively, a gradient of expression of Sox9 and Lhx2 observed in developing hair follicles leads to a few Sox9 and Lhx2 double positive cells at the transitional zone²², and might thus result in labelling of Merkel cells in the Lhx2-CreER lineage tracing experiment."

2. In Fig. 1c and d, Atoh1+Merkel cells lack Sox9 at E14.5 and E15.5. Nonetheless, the lineage tracing of Sox9 cells shows Sox9+cells give rise to Merkel cells. It suggests that Sox9 expression in Merkel cell progenitor is very transient and the authors can analyze or mention this.

We analysed if Atoh1(+) Merkel cells are also Sox9(+) by performing IF staining and could not detect any Sox9/Atoh1 co-labelling. Thus, it is likely that once the Merkel cells are specified, the expression of Sox9 is downregulated in these cells. We included images to illustrate this point in Supplementary Figure 1a-c' and included the following sentence in the Results section: "Regardless of cell proximity, Atoh1-GFP(+) Merkel cells did not co-label with Sox9 (Supplementary Fig. 1a-c) or Lhx2 (Fig. 1f, g)." We also added the following to the Discussion section: "While Sox9(+) cells give rise to Merkel cells, our data show that the Sox9 protein itself is absent from Atoh1(+) Merkel cells and Sox9 is not required for Merkel cell specification."

3. Sox9+cells are absent in Smo-null skin epithelium at P0. Can the authors confirm the Sox9 expression at earlier than P0 (e.g. E13.5) given sox9 expression may be transient in wild type skin?

Unfortunately, we had difficulties obtaining E13.5 embryos, as we do not have a running colony of Smo cKO mice. We did have samples from E16 Smo cKO mice and performed immunofluorescence analysis of Sox9 in these animals. We did not observe any Sox9(+) cells. We included these data in the revised manuscript, which now reads as follows: "Importantly, immunofluorescence analysis of Sox9 showed that there was an absence of Sox9(+) cells in embryonic and neonatal hair follicles of Smo-null skin, whereas these cells were apparent in control hair follicles (Supplementary Fig. 2d, e)".

4. Is Fgfr2 expression confined to Merkel cell progenitor?

No, Fgfr2 expression is not confined to Merkel cell progenitors and is present in the epidermis, as shown in Supplementary Figure 4f and 4g, as well as Supplementary Figure 4h and 4i in control skins.

5. *Is Sox9+ cells detected in Fgf20-null skin like Fgfr2-null skin? In addition, it would be more convincing if quantifications of Sox9+ cells in Fgf20- and Fgfr2-null skin are added.*

The number of Sox9(+) cells is reduced in Fgf20-null skins, but these cells are still detected. It is known that during primary hair formation, loss of Fgf20 affects both the epithelium and the mesenchyme (PMID:23431057), and we can not exclude an indirect role of Fgf20 in controlling the number of Sox9(+) cells. In order not to confuse the readers, we decided not to include these data.

The more relevant finding is that epithelial Fgfr2-mediated signalling functions to control Merkel cell specification. Thus, we quantified the number Sox9(+) cells in control and Fgfr2 cKO skins and did not observe any statistically significant differences (Figure 5e). This indicates that the drastic reduction in the number of Merkel cells observed in Fgfr2 cKO skins is not due to changes in the number of Sox9(+) cells. We added the following sentence to the text describing these data: “Furthermore, immunofluorescence analysis also showed that Sox9(+) cells were present in both control and Fgfr2-null hair follicles (Fig. 5d-f).”

6. *Does Fgf20 act in autocrine manner in Fgfr2+ cells? or do other cells type secrete Fgf20?*

We performed immunofluorescence analysis of Fgfr2 in Fgf20-LacZ skin and found a gradient of Fgf20 expression in the Fgfr2(+) cells. While some Fgfr2(+) cells had low levels of Fgf20, others had almost no Fgf20 expression. Thus, one can not conclude whether Fgf20 functions in an autocrine or paracrine manner in Fgfr2(+) cells. We included these data in Supplementary Figure 4h. We added the following sentence to the text describing these data: “Furthermore, by analysing the skins of Fgf20-LacZ mice, we observed a gradient in the expression of Fgf20 in Fgfr2(+) and Sox9(+) cells, where some Fgfr2(+) and Sox9(+) cells had some levels of Fgf20, while others had none (Supplementary Fig. 4h, i).”

7. *Does Sox9 repress Fgf20 or in the cells giving rise to Merkel cells?*

We performed immunofluorescence analysis of Sox9 in Fgf20-LacZ skins and found different levels of Fgf20 between Sox9(+) cells. Some Sox9(+) cells had low levels of Fgf20, and these cells were mainly located at the caudal side of the hair follicles. On the other hand, Sox9(+) cells at the leading edge of the hair follicles had strong levels of Fgf20. Thus, while our data clearly show that Sox9(+) cells give rise to Merkel cells, one can not necessary conclude that Sox9 represses Fgf20 in these cells. We included these data in Supplementary Figure 4i. We added the following sentence to the text describing these data: “Furthermore, by analysing the skins of E16 Fgf20-LacZ mice, we observed a gradient in the expression of Fgf20 in Fgfr2(+) and Sox9(+) cells, where some Fgfr2(+) and Sox9(+) cells had some levels of Fgf20, while others had none (Supplementary Fig. 4h, i).”

8. *Labels are missing; Fig. 4g and Fig. 5h*

We double-checked the labels to make sure that they are in place.

REVIEWERS' COMMENTS:

Reviewer #1 (Remarks to the Author):

The authors have provided excellent supporting data for their claims and have adequately added to their discussion and manuscript to address previous concerns. We support the publication of this manuscript, as it clarifies and enhances our understanding of Merkel-cell development.

Reviewer #2 (Remarks to the Author):

The authors have submitted a revised and improved manuscript that addresses many reviewer comments, however the following comments remain.

1. In their response to comments, the authors agree that their data supports a heterogeneous population of Sox9 cells in the developing skin with differing biology and fate potential. Parts of the revised manuscript embrace this as well. However there remain other parts of the text that directly imply that the same Sox9 cells fated to form hair follicle also form Merkel cells. For example: in the abstract "Merkel cells originate from Sox9 positive (+) cells inside hair follicles, which were previously known to give rise to hair follicle stem cells (HFSCs) and cells of the hair follicle lineage." In the introduction "we discovered that Merkel cells originate from Sox9(+) cells, which give rise to cells of the hair follicle lineage...". Consider revising to avoid confusion about your study's conclusions.
2. In the revised sentence: "These activities are largely provided by the epidermis, hair follicles, and Merkel cells, which serve protective barrier functions, provide thermoprotection, and are involved in mechanosensation, respectively." This implies that skin sensory functions are "largely provided by" Merkel cells. This is not accurate as skin somatosensation uses multiple sensory nerve types and mice deficient for Merkel cells have minimal gross sensory defect.
3. In the revised sentence: "Fibroblast growth factor (Fgf) 20 is expressed by hair placodes (Supplementary Fig. 4a, b) and is required for primary and secondary hair follicle formation." This still misrepresents the data from the cited paper. Although there are changes in the frequency of secondary hair shaft types, the secondary follicles form in a relatively normal way. In contrast, the Fgf20 mutant primary follicles do not develop fully.
4. In the sentence: "Thus, we concluded that hair follicle development growth rate in Fgfr2 cKO is not slower compared to the control rate." The growth rate was not really measured (difference in growth between two points in time). Moreover, attaining a shorter length over the same amount of time does imply a growth defect. Perhaps the authors are not trying to make conclusion about developmental growth rates from their data?
5. In the conclusion: "...we can conclude that Merkel cell progenitors are Sox9(+) Shh(-) Lhx2(-) cells." It would be more accurate to say they are predominantly these cells.
6. In the conclusion: "basement layer" of the epidermis should be "basal layer".
7. In the conclusion: "Xiao and colleagues observed no changes in Merkel cell numbers in P0 Fgf20 KO mice compared to controls." The cited work does report a reduction in the number of Merkel cells, but a smaller reduction than is found in the present work.

Reviewer #3 (Remarks to the Author):

The authors addressed my comments from the previous review. I have no further comments.

Reviewer #2 (Remarks to the Author):

The authors have submitted a revised and improved manuscript that addresses many reviewer comments, however the following comments remain.

1. In their response to comments, the authors agree that their data supports a heterogeneous population of Sox9 cells in the developing skin with differing biology and fate potential. Parts of the revised manuscript embrace this as well. However there remain other parts of the text that directly imply that the same Sox9 cells fated to form hair follicle also form Merkel cells. For example: in the abstract “Merkel cells originate from Sox9 positive (+) cells inside hair follicles, which were previously known to give rise to hair follicle stem cells (HFSCs) and cells of the hair follicle lineage.” In the introduction “we discovered that Merkel cells originate from Sox9(+) cells, which give rise to cells of the hair follicle lineage...”. Consider revising to avoid confusion about your study’s conclusions.

To avoid implying that the same Sox9(+) cells fated to form HFs also form Merkel cells, we have made the following change in the abstract: “Sox9 positive (+) cells inside hair follicles, which were previously known to give rise to hair follicle stem cells (HFSCs) and cells of the hair follicle lineage, can also give rise to Merkel Cells”.

We have also made the following change in the introduction “, we discovered that Sox9(+) cells, which in prior literature have been proven to give rise to cells of the hair follicle lineage, including HFSCs that maintain postnatal hair follicle growth and homeostasis, can also give rise to Merkel Cells”.

2. In the revised sentence: “These activities are largely provided by the epidermis, hair follicles, and Merkel cells, which serve protective barrier functions, provide thermoprotection, and are involved in mechanosensation, respectively.” This implies that skin sensory functions are “largely provided by” Merkel cells. This is not accurate as skin somatosensation uses multiple sensory nerve types and mice deficient for Merkel cells have minimal gross sensory defect.

We have modified this part as suggested by reviewer #2. The modified text now reads as follows “ The skin epithelium is an essential barrier that protects the body from the environment, helps to maintain temperature and keep water within the body, and performs sensory functions¹. These activities are largely provided by the epidermis, hair follicles, and specialized cells, including Merkel cells, which respectively serve protective barrier functions, provide thermoprotection, and are involved in mechanosensation¹⁻³ .

3. *In the revised sentence: “Fibroblast growth factor (Fgf) 20 is expressed by hair placodes (Supplementary Fig. 4a, b) and is required for primary and secondary hair follicle formation.” This still misrepresents the data from the cited paper. Although there are changes in the frequency of secondary hair shaft types, the secondary follicles form in a relatively normal way. In contrast, the Fgf20 mutant primary follicles do not develop fully.*

We have modified this sentence to avoid misrepresenting the data from cited paper, and it now reads follows “Fibroblast growth factor (Fgf) 20 is expressed by hair placodes (Supplementary Fig. 4a, b) and is required for the proper development of primary hair follicles³¹”.

4. *In the sentence: “Thus, we concluded that hair follicle development growth rate in Fgfr2 cKO is not slower compared to the control rate.” The growth rate was not really measured (difference in growth between two points in time). Moreover, attaining a shorter length over the same amount of time does imply a growth defect. Perhaps the authors are not trying to make conclusion about developmental growth rates from their data?*

We have modified this sentence accordingly, without making conclusions about the developmental growth rates of Fgfr2 cKO HFs, and now it reads as follows “Thus, we concluded that the development and cell differentiation were not arrested in Fgfr2 cKO hair follicles”.

5. *In the conclusion: “...we can conclude that Merkel cell progenitors are Sox9(+) Shh(-) Lhx2(-) cells.” It would be more accurate to say they are predominantly these cells.*

We thank Reviewer#2 for this comment. The text and now reads as follows “Taking our study and the published data together, we can conclude that Merkel cell progenitors are predominantly - Sox9(+) Shh(-) Lhx2(-) cells. “

6. *In the conclusion: “basement layer” of the epidermis should be “basal layer”.*

We have modified the text and it now reads as follow “Furthermore, during development, hair follicles grow downward into the underlying dermis, while Merkel cells are located upwards in the basal layer of the epidermis”

7. *In the conclusion: “Xiao and colleagues observed no changes in Merkel cell numbers in P0 Fgf20 KO mice compared to controls.” The cited work does report a reduction in the number of Merkel cells, but a smaller reduction than is found in the present work.*

We have modified the text and it now reads as follow “Recently, Xiao and colleagues²⁴ observed a drastic reduction in the expression of a key Merkel cell differentiation factor, Atoh1,

in E15.5 Fgf20 KO compared to control. Interestingly, however, despite the reduction in *Atoh1* expression at the time of Merkel cell specification, Xiao and colleagues²⁴ observed a small reduction in Merkel cell numbers in P0 Fgf20 KO mice compared to controls. While this result is different to our findings, this discrepancy could be due to differences in mouse genetic backgrounds used in two studies and thus, for some mouse genetic backgrounds lacking Fgf20, other Fgf ligands can sufficiently compensate for the loss of Fgf20 and specify Merkel cells.”